# Advanced Paternal Age: A New Indicator for the Use of Microfluidic Devices for Sperm DNA Fragmentation Selection

**DOI:** 10.3390/jcm13020457

**Published:** 2024-01-14

**Authors:** Laura Escudé-Logares, Clara Serrano-Novillo, Laia Uroz, Anna Galindo, Carmen Márquez

**Affiliations:** Gravida, Hospital de Barcelona, 08034 Barcelona, Spain; lescude@gravidabcn.com (L.E.-L.); cmarquez@gravidabcn.com (C.M.)

**Keywords:** microfluidic sperm selection, sperm DNA fragmentation, advanced paternal age, ICSI, dsSDF

## Abstract

New social conditions and progress in ART have both contributed to the delay in parenthood in developed countries. While the effects of maternal age have been widely studied, paternal age is poorly understood, and there are no specific guides on ART techniques to treat its deleterious effects. It is known that there is an increase in sperm DNA fragmentation (SDF) in elderly men, and new sperm selection devices using microfluids have been developed. This study analyses 189 ICSI cycles with donor oocytes performed between January 2018 and February 2022. Spermatozoa were selected using an MSS device or density gradients, followed by ICSI fertilization and fresh/thawed embryo transfer. We assessed the association between the selection technique, paternal age (< or ≥45) and reproductive outcomes. Fertilization (FR), blastulation (BR), implantation (IR), live-birth (LBR) and miscarriage (MR) rates were calculated. The results showed significantly higher IR (57.7% vs. 42.5%) and LBR (42.9% vs. 30.3%) when applying MSS selection, and particularly higher BR, IR and LBR when the paternal age was equal to or over 45 years (BR: 64.4 ± 23% vs. 50.1 ± 25%, IR: 51.5% vs. 31.6% and LBR: 42.4% vs. 23.7%). We also found a negative correlation between BR and paternal age (r2 = 0.084). The findings show that MSS enhances success in assisted reproduction cycles with ICSI, especially in couples with advanced paternal age. We propose advanced paternal age as a new indicator for the application of sperm selection techniques that reduce fragmentation.

## 1. Introduction

It is known that paternal age correlates with worst sperm parameters and reproductive outcomes. Sperm count, motility, morphology or viability, among others, are negatively affected in elderly men. Many potential causing mechanisms associated with age have been described, including the decreased function of reproductive accessory glands or changes in male reproductive anatomy [1]. Studies on advanced paternal age (APA) have shown a significant increase in sperm aneuploidies, although its implication in embryo aneuploidies remains unclear [2,3]. Associations with congenital birth defects and compromised DNA repair mechanisms have also been described [4], and several studies have shown an increase in sperm DNA fragmentation (SDF) in elderly men, considered men aged 40–50 or above [5,6,7,8].

Sperm genome damage, whether caused by an abnormal chromatin composition, protamine deficiency or SDF, is linked to male infertility [9]. The study of SDF has aroused great interest among the scientific community in the last few years. There are several causes of SDF: apoptotic processes, non-repaired breaks occurring during meiotic recombination, the histone-to-protamine replacement process or oxidative stress caused by endogenous or exogenous processes, among others [7,10,11]. SDF can be classified according to whether it affects one strand of DNA (single-strand sperm DNA fragmentation (ssSDF)) or both strands (double-strand sperm DNA fragmentation (dsSDF)). Double-strand SDF can be subdivided into extensive dsSDF and dsSDF localized in MAR regions. These regions correspond to around 15% of human sperm chromatin, which remains packed with histones and links toroids to each other and to the nuclear matrix. These regions, which contain coding and regulatory sequences, are more accessible to damage [12,13,14].

These different types of SDF lead to different consequences for reproduction. Single-strand SDF is mainly due to oxidative stress and has been associated with lower pregnancy rates. Extensive dsSDF would probably be lethal for the zygote since no complementary strand would be available for DNA repair as paternal and maternal pronuclei remain separated in early mammalian embryos. The dsSDF located in MAR regions, caused by an increase in breaks during meiotic recombination or a defect in the repair pathways [15], remains attached to the nuclear matrix, which is part of the male pronucleus, and is not disorganized until the first mitotic division. This fact could allow the correct repair of the breaks at this stage of development, which plays a key role in oocyte quality. However, when not repaired or misrepaired, these breaks can cause a delay in the cell cycle and, therefore, in embryonic development through blastocyst, and lead to implantation failures and an increase in the rate of miscarriages [12,16,17,18].

Currently, there are several techniques to study SDF [19]: the sperm chromatin dispersion test (SCD), the sperm chromatin structure assay (SCSA), the terminal transferase dUTP nick end labelling (TUNEL) and gel electrophoresis (COMET). Only COMET allows differentiation between ssSDF and dsSDF through alkaline and neutral pH conditions, respectively [16,19]. SCD, SCSA and TUNEL offer a single SDF value, regardless of the type. In addition, neutral COMET is the only technique that would allow to mostly detect the dsSDF localized in MAR regions, while SCD, SCSA and TUNEL could detect the extensive dsSDF but cannot distinguish it from ssSDF [12].

Conventional sperm evaluation only involves the microscopic study of the sample, measuring sperm concentration, motility and morphology [20]. However, studies show that samples classified as normal by these parameters may carry aneuploidies or a high rate of SDF [21,22]. This is important in cases where assisted reproductive technologies (ART) are applied, especially in cycles with intracytoplasmic sperm injection (ICSI), where a single sperm with good motility and morphology is selected to fertilize the oocyte [15].

To select those spermatozoa of higher quality with a lower rate of SDF, microfluidic sperm selection (MSS) devices have been developed. ZyMōt^TM^ICSI (Koek Biotechnology, Izmir, Türkiye), applied in this study, mimics the conditions of the female genital tract. Sperm selected by this method could show lower SDF and reactive oxygen species rates and better morphology and motility (Cimab Ibérica, s.d.).

In this retrospective study, we analyse the effects of the application of an MSS device on the outcomes of ART cycles with donor oocytes and ICSI fertilization. We assess whether there is a significant improvement in fertilization (FR), blastulation (BR), implantation (IR), live-birth (LBR) and miscarriage (MR) rates when applying MSS by ZyMōt^TM^ICSI compared to selection by density gradients in different cohorts according to paternal age. We analyse the effect of APA and whether the application of MSS could modulate the early outcomes. New social conditions and progress in ART have both contributed to the delay in parenthood in developed countries. While the effects of maternal age have been widely studied, paternal age is poorly understood, and there are no specific guides on ART techniques to treat its deleterious effects. With our results, we propose APA as an important diagnostic criteria to consider for the use of MSS to improve ART outcomes.

## 2. Materials and Methods

### 2.1. Study Design

This retrospective study was conducted at Gravida Fertilitat Avançada in Barcelona, Spain. The present cohort study was drawn from a total of 179 couples undergoing 189 treatments with donor oocytes fertilized using intracytoplasmic sperm injection (ICSI) performed between January 2018 and February 2022. MSS was applied using the ZyMōt^TM^ICSI device in 102 treatments presenting an altered fragmentation test or following a medical assessment (including patients with a reproductive history of bad blastulation, poor embryo quality with extended culture, miscarriages or implantation failure, following medical recommendations based on each particular case). In the other 87 cases, sperm selection was performed using density gradients (non-MSS). Cycles with frozen sperm samples and sperm obtained by a testicular biopsy were excluded. The advanced paternal age threshold was established at 45 years of age. Treatments were divided according to whether paternal age was below 45 years (n = 132) or equal to or above (n = 57).

All patients’ data were collected as part of the routine clinical procedure and remain confidential, and the patient’s anonymity is maintained. Therefore, this observational study has no risk of compromising the identity or safety of the individual. Owing to its retrospective nature and absence of a priori intervention, this study was exempt from external review board approval.

### 2.2. Ovarian Stimulation

To determine the ovarian stimulation protocol, the patient’s age, antral follicle count, hormone levels and prior treatments were considered. Recombinant alpha or beta FSH (with or without urinary menotropins) ranging from 150 to 375 IU was used for ovarian stimulation ((Gonal F^®^ (Merck, Darmstadt, Germany) or Menopur^®^ (Ferring, Kastrup, Denmark)). Then, from day 5 or 6 after FSH injections, GnRH antagonists (Orgalutran^®^ (Organon, Amsterdam, The Netherlands)) were administered daily to prevent ovulation prior to egg collection. Finally, hCG and/or a GnRH agonist was used to trigger final follicular maturation (Ovitrelle^®^ (Merck, Darmstadt, Germany) and Decapeptyl^®^ (Ipsen, Boulogne-Billancourt, France), respectively). A GnRH agonist was used when patients had ≥10 follicles over 14 mm or estradiol levels ≥ 2500 pg/mL. If lower levels or follicles were found, a combination of hCG and GnRH was administered (250 µg and 0.2 mg, respectively).

### 2.3. Oocyte Retrieval

Oocyte retrieval was performed using a transvaginal needle under ultrasound guidance 36 h after the triggering of follicular maturation, under the sedation of the patients. Oocytes were incubated in fertilization medium (Global for Fertilization^®^, LifeGlobal, Guilford, CT, USA) under an oil (Ovoil™, Vitrolife, Gothenburg, Sweden) overlay at 37 °C and 6.6% CO_2_ with 5% O_2_ (previously equilibrated overnight). Hyaluronidase (HYASE™, Vitrolife) and pipetting were used to remove the cumulus cells surrounding the oocytes, and the maturation status of the oocytes was assessed. Only metaphase II (MII) oocytes were prepared for ICSI.

For vitrified donor oocytes, oocytes were devitrified according to the manufacturer’s instructions (Kitazato, Shizuoka, Japan).

### 2.4. Semen Preparation

Semen was left to liquify for at least 20 min at room temperature. Density gradients were prepared by adding 1 mL of the 90%, 70% and 50% gradients. Semen samples were layered on the gradient tube and centrifuged at 250× *g* for 10 min. The recovered pellet was resuspended in 2 mL of medium and centrifuged twice for 3 min at 450× *g*. Finally, a processed sample for ICSI was recovered.

For the microfluidic technique, ZyMōt^TM^ICSI was used. ZyMōt^TM^ICSI is a single-use microfluidic device (76.4 mm × 24.5 mm × 1.76 mm) with 5 inlet chambers for sperm loading, followed by capacitation chambers that will allow the flow of sperm with reduced dsSDF. Each inlet and outlet port of the ZyMōt^TM^ICSI was filled with 6.5 μL of medium. Next, 2 μL of the semen sample were added to each channel inlet (one device for each patient), followed by 6.5 μL of mineral oil in both the inlet and outlet ports. Then, the microfluidic chip was placed on a 37 °C heated surface until enough spermatozoa reached the outlet channel within a maximum of 30 min. Spermatozoa were collected from the outlet ports, and those with better morphology and motility were selected for ICSI fertilization.

### 2.5. ICSI and Embryo Culture

MII oocytes were microinjected with the spermatozoa sorted by conventional density gradients or by ZyMōt^TM^ICSI and then placed on individual microwells of time-lapse incubators or conventional culture plates, depending on the case. Fertilization was checked 16–18 h post-insemination. The presence of two pronuclei and two polar bodies ensured normally fertilized oocytes, and the fertilization rate was calculated for each patient as the number of normally fertilized oocytes among MII oocytes. Embryo culture was performed in medium (Global Total LP^®^ medium (Vitrolife)) covered in mineral oil (Ovoil^TM^, Vitrolife) for up to 6 days at 37 °C with 6–6.7% CO_2_ and 5% O_2_. Embryo grading was performed following Alikani et. al. recommendations [23], Gardner classification [24] and ESHRE and ASRM guidance [25,26]. In those treatments where embryo culture was brought to blastocyst, the blastulation rate was calculated for each patient (n = 163).

### 2.6. Embryo Transfer and Clinical Outcomes

Embryo transfer, performed under ultrasound guidance, was carried out on day 3 or 5 of embryonic development and one or two embryos were transferred, depending on medical criteria or the patients’ choice. Both cycles with fresh embryo transfer (n = 98) and cycles with thawed embryo transfer (n = 210) were considered to analyse the implantation rate, calculated as the number of gestational sacs per number of transferred embryos (n = 361). The live birth rate was calculated as the number of live births per number of embryos transferred, and the miscarriage rate was the result of the number of spontaneous abortions per number of positive blood pregnancy tests (n = 181).

### 2.7. Statistical Analysis

Statistical analyses were performed using the R Commander graphical user interface (R software 4.0.2 version). Normalization of the data was determined with Shapiro–Wilk (n < 50) or Kolmogorov–Smirnov (n > 50) tests. The means of parametric samples were computed using an independent sample T-test, and non-parametric samples were compared using unpaired two-samples Wilcoxon test. The difference in proportions was analysed using the Chi-square test of independence or the Fisher exact test for cases where the expected frequencies were below five. The correlation was determined using the Pearson correlation test, and a linear regression model was performed. *p* ≤ 0.05 was considered significant.

## 3. Results

A total of 179 couples undergoing 189 treatments with donor oocytes fertilized using intracytoplasmic sperm injections performed between January 2018 and February 2022 were analysed.

There were no statistical differences in the age of receptor women nor in paternal age between MSS and non-MSS groups. The statistical study did not show significant differences in the FR, BR or MR, but it did show significant differences in the IR and LBR of the cycles in which ZyMōt^TM^ICSI selection was applied compared to the non-MSS group. The mean IR of the MSS group was 57.7%, while that of the non-MSS group was 42.5% (*p*-value = 0.004). The mean LBR of the MSS group was 42.9%, while that of the non-MSS group was 30.3% (*p*-value = 0.014) (Table 1).

The scatter plot obtained considering only those patients in whom ZyMōt^TM^ICSI had not been applied showed a negative trend between paternal age and BR (Figure 1a). This resulted in a significant negative correlation (r = −0.314, *p*-value = 0.012) (Table 2). The linear regression model provided an r^2^ = 0.099 and a significant *p*-value of 0.012 (Figure 1a). No correlation was detected between paternal age and FR (Table 2).

In contrast, we did not find any trend between paternal age and BR when ZyMōt^TM^ICSI was applied (Figure 1b). The correlation test also showed no significant correlation between these two variables (*p*-value = 0.413) or between paternal age and FR (Table 2).

When focusing on male patients under 45 years of age, we found no significant differences in any of the rates analysed between cycles with MSS and without MSS, although there was a notable improvement in IR and LBR.

In contrast, patients ≥ 45 years of age presented a significant effect on the BR, IR and LBR when applying MSS. The mean BR of the MSS group was 64.4 ± 23%, while that of the non-MSS group was 50.1 ± 25% (*p*-value = 0.034). The mean IR of the MSS group was 51.5%, and for the non-MSS group, it was 31.6% (*p*-value = 0.048). The mean LBR of the MSS group was 42.4%, while that of the non-MSS group was 23.7% (*p*-value = 0.048). There were no significant differences in FR or MR (Table 3).

## 4. Discussion

This study assessed the effect of applying MSS using ZyMōt^TM^ICSI in ART cycles with donor oocytes and ICSI fertilization, evaluating the impact on fertilization, blastulation, implantation, live-birth and miscarriage rates. Our results showed significantly higher IR and LBR when applying MSS and significantly higher BR, IR and LBR when applying MSS in cycles where the paternal age was equal to or above 45 years.

In recent years, the effects of SDF on outcomes following ART cycles with ICSI have been controversial, with some studies failing to find any association between SDF and outcomes in ICSI cycles [11,27] and others showing worse outcomes for SDF patients [28,29]. This could be a consequence of the inclusion and exclusion criteria established, the fragmentation threshold value considered pathological or the lack of distinction between ssSDF and dsSDF [12,15].

In the present study, we have found an improvement in IR and LBR. The average IR and LBR achieved by applying MSS were 15.2% and 12.6% higher than with density gradient selection, respectively. This implicitly involves a decrease in the number of miscarriages and/or implantation failures. However, no significant differences in MR have been reported in this study. Yet, we cannot dismiss the fact that the receptor had other factors that could mask the effects of selection by MSS, such as immunological factors, endometrial microbiota or uterus morphology [30,31,32]. FR has been described as being mainly related to ssSDF [15,33,34]. Previous studies have found a negative correlation between ssSDF and progressive motility [5,12,15]. Given that sperm selection during an ICSI procedure is based on good motility and morphology, spermatozoa selected for ICSI have significantly lower levels of ssSDF in comparison to spermatozoa in the ejaculate. By performing ICSI, we suppress the differences between both study groups without the need for MSS. Accordingly, no differences were found in the application of MSS to FR. Interestingly, in contrast to ssSDF, no differences have been found in dsSDF values between spermatozoa selected for ICSI and those from ejaculate, so sperm selection for ICSI does not reduce the rates of dsSDF located in MAR regions [15]. Regarding BR, the proportion of good-quality embryos was higher in the MSS group (61% vs. 57.5%) but not statistically significant. In this sense, other authors have proven an improvement in BR when applying microfluidic devices [35]. However, in our study, we did find significant differences in BR when populations were divided by paternal age. This suggests that having the subgroups mixed in a general study group may mask the effect of the MSS and reveal that a specific group of patients may be the indicated group for its application.

Our correlation studies in cycles without MSS showed a negative correlation between BR and paternal age. In accordance, the mean BR in older men was 12.3% lower than in men under 45 years in cycles without MSS. These findings coincide with previous studies showing that SDF increases with age and worsens ART outcomes [8,36]. It has been observed that in older men, there is greater sperm genome fragility, making it more vulnerable to damage caused by endogenous or exogenous factors [7]. In fact, it has been demonstrated that, with age, there are changes in the gene expression of protamine genes, transcription factors and miRNAs expression patterns in seminal plasma. This leads to an alteration in the appropriate ratio between protamine 1 and 2 (P1/P2), making the genome more accessible to nuclease action [7]. If the repair capacity of the oocyte is exceeded, fragmentation may be irreversible [7,15,37]. Moreover, several studies have shown a correlation between male age and sperm aneuploidies [2,3], which could worsen BR, LBR or MR, among others. The use of donor oocytes in this study eradicated any difference in their repair capacity due to maternal age. When applying MSS, the correlation disappeared, obtaining a similar BR in all ages studied. In fact, significant differences in BR only appeared when the age was equal to or higher than 45 years, increasing BR by 14.3%. In this subgroup, the improvement in IR and LBR was even more pronounced, improving the IR by 19.9% and the LBR by 18.7%.

This evidence demonstrate that advanced paternal age worsens ART outcomes (probably by an increase in SDF) and that selection by MSS is an effective method to reduce these effects. Therefore, advanced paternal age, in this study considered to be 45 years or older based on previous studies [5,6,7,8], could be a sufficient factor to indicate the use of ZyMōt^TM^ICSI without the need to perform a previous fragmentation test. We demonstrate an improvement in outcomes for APA patients by applying MSS, both following COMET test results and medical criteria based on couples’ reproductive histories. Moreover, older men commonly tend to have older female partners, carrying the worst capacity to repair DNA damage. If cycles with our own oocytes were included in the study, we would expect even higher improvements when applying MSS to the >45 years old male group, but it still needs to be tested.

Previous studies have shown a delay in embryo kinetics, worse IR and an increase in MR in ICSI cycles with semen samples with a high rate of dsSDF but not with a high rate of ssSDF [16,17,38]. One plausible explanation, besides the low ssSDF selection by motility with ICSI per se [15], is that ssSDF can be effectively repaired using the complementary strand as a template prior to DNA replication, while unrepaired or misrepaired dsSDF can lead to chromosomal aberrations triggering the consequences mentioned above [16]. The improvement we found in BR, IR and LBR when applying ZyMōt^TM^ICSI in ICSI cycles with oocyte donation can be added to the fact that extensive dsSDF may be lethal to the zygote. All these facts support dsSDF located in MAR regions as the most frequent type of fragmentation in spermatozoa selected and microinjected for ICSI [15]. We support the data that ZyMōt^TM^ICSI is an effective device for reducing the effects of dsSDF in ART cycles with ICSI.

Our results are in accordance with the only previous study found in the literature evaluating the impact of ZyMōt^TM^ICSI selection in comparison with Swim-Up, which showed a higher proportion of good-quality embryos when applying MSS [35]. Yetkinel’s study also lacked previous fragmentation tests and randomised patients into the MSS or Swim-Up groups [35]. In our case, given the lack of fragmentation tests in some couples, the application of ZyMōt^TM^ICSI was empirical in those cases, generally related to previous cycles with bad blastulation rates and good oocyte quality (commonly associated with male factor and DNA fragmentation [16,17,38]). Therefore, there could be patients where MSS was applied without having SDF and others with high SDF rates where this selection system was not performed. This supports the assumption that in both populations, there are patients with high SDF rates and, thus, that the populations are homogeneous and allow the detection of an improvement when applying MSS. Our findings are also in line with previous studies using other MSS devices, which have reported an improvement in embryo quality as well as in implantation and pregnancy rates [39,40].

The results of this study have shown that SDF causes significant detrimental effects on ART cycles with ICSI, mainly affecting BR, IR and LBR, which decrease significantly and reduce reproductive success despite the existence of good oocyte quality. The MSS device ZyMōt^TM^ICSI has been validated as an effective system for reducing these effects and increasing success rates, especially for aged male patients. However, while our study shows a clear improvement in outcomes when using the device in older patients, it could be interesting to carry out future studies with a larger population and potentially include autologous ICSI cycles with previous neutral COMET fragmentation tests to confirm the increase in dsSDF with male age and its reduction when applying MSS. These future studies should have stricter inclusion and exclusion criteria, be in possession of previous fragmentation tests using the neutral COMET technique (to avoid the possible bias of populations studied when MSS vs. non-MSS groups are determined in the absence of these tests, as is the scenario in the present study), and have a larger population to give greater statistical power to the results.

## 5. Conclusions

Microfluidic sperm selection enhances success in assisted reproduction cycles with ICSI, increasing the rate of implantation and live births. It is particularly important in couples with advanced paternal age, where the rate of blastulation is also significantly improved. Therefore, MSS devices enable them to reduce the effects of APA, making APA a potential indicator for their application despite the fragmentation test result. While this conclusion must be further confirmed with a prospective randomized study and a larger sample size, our results suggest it could be interesting to recommend the use of MSS devices for males equal to or over 45 years old.

## Figures and Tables

**Figure 1 jcm-13-00457-f001:**
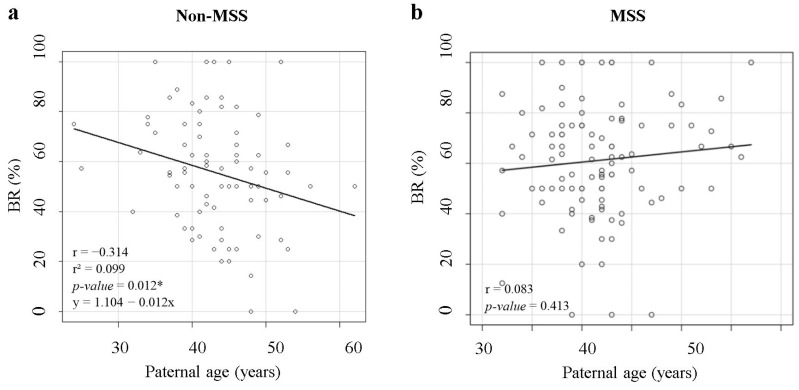
BR in relation to paternal age. (**a**) Scatter plot of the BR in relation to paternal age in cycles without MSS. A summary of the linear regression model supporting the correlation is shown in the lower left corner. r: coefficient of correlation; r^2^: coefficient of determination; * *p*-value < 0.05. The regression line is shown in the lower row. (**b**) Scatter plot of the BR in relation to paternal age in cycles with MSS. MSS: microfluidic sperm selection; BR: blastulation rate.

**Table 1 jcm-13-00457-t001:** Comparison of clinical outcomes of cycles with MSS vs. without MSS.

Variables	Non-MSS		MSS		*p*-Value
	n	M ± SD	n	M ± SD	
Receptor age	87	42.7 ± 4.0	102	41.7 ± 3.3	NS
Paternal age	87	42.7 ± 6.4	102	41.7 ± 5.3	NS
	n	%	n	%	
FR (M ± SD)	87	71.7 ± 16	102	75.0 ± 18	NS
BR (M ± SD)	63	57.5 ± 24	100	61.0 ± 22	NS
IR	179	42.5	182	57.7	0.004 **
LBR	178	30.3	177	42.9	0.014 *
MR	78	34.6	103	33	NS

MSS: microfluidic sperm selection; FR: fertilization rate; BR: blastulation rate; IR: implantation rate; LBR: live-birth rate; MR: miscarriage rate; * *p*-value < 0.05; ** *p*-value < 0.01; M: mean; SD: standard deviation; n: sample size; NS: non-significant.

**Table 2 jcm-13-00457-t002:** Correlation study of the FR and BR with paternal age in cycles without MSS vs. with MSS.

	Variables	n	Correlation	*p*-Value
Non-MSS	FR	87	−0.006	NS
	BR	63	−0.314	0.012 *
MSS	FR	102	−0.118	NS
	BR	100	0.083	NS

MSS: microfluidic sperm Selection; FR: fertilization rate; BR: blastulation rate; * *p*-value < 0.05; n: sample size; NS: non-significant.

**Table 3 jcm-13-00457-t003:** Comparison of clinical outcomes of cycles with MSS vs. without MSS in men under 45 years of age vs. men aged 45 and over.

	Variables	Non-MSS		MSS		*p*-Value
Paternal Age		n	%	n	%	
<45 years	FR (M ± SD)	51	71.7 ± 15	81	76.5 ± 18	NS
	BR (M ± SD)	38	62.4 ± 22	79	60.2 ± 22	NS
	IR	103	50.5	149	59.1	NS
	LBR	102	35.3	144	43.1	NS
	MR	50	34	87	33.3	NS
≥45 years	FR (M ± SD)	36	71.7 ± 18	21	69.3 ± 15	NS
	BR (M ± SD)	25	50.1 ± 25	21	64.4 ± 23	0.034 *
	IR	76	31.6	33	51.5	0.048 *
	LBR	76	23.7	33	42.4	0.048 *
	MR	28	35.7	16	31.2	NS

MSS: microfluidic sperm selection; FR: fertilization rate; BR: blastulation rate; IR: implantation rate; LBR: live-birth rate; MR: miscarriage rate; * *p*-value < 0.05; M: mean; SD: standard deviation; n: sample size; NS: non-significant.

## Data Availability

Data are contained within the article.

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
