# Peer review of "Advanced Paternal Age: A New Indicator for the Use of Microfluidic Devices for Sperm DNA Fragmentation Selection"

_jcm, 2024, doi:10.3390/jcm13020457_

Round 1
Reviewer 1 Report
Comments and Suggestions for Authors
The authors had conducted a very meaningful study, which has certain guiding significance for improving the success rate of ICSI in clinic. It should be pointed out that the study was conducted retrospectively and the evidence is of average strength. If high-quality prospective studies can be carried out later, the results will have stronger support. Authors need to pay attention to the wording of the results in the manuscript.
Reviewer 2 Report
Comments and Suggestions for Authors
Reviewer 3 Report
Comments and Suggestions for Authors
Dear authors, congratulations for your hard work. It is true, that advanced paternal age is on eof the main risk factors associated with male infertility.
If I may I want to share some thoughts with you. It appears to me that your paper is in favour of a specific technique and therefore you should change the title of it. You haven't described any other techniques and your data are driven by the implementation or not of that specific technique. This isn't wrong if you are trying to validate or enhance the strength and value of it. But you cannot generalize and use plural , meaning techniques, in your title.
Having that said let's get a deep look at your work. It is a retrospective one while it would be better to have a prospective one.
You do not specify what made you use density gradients in 87 cases (DFI <30%, paternal age <45, what?) Samewise you do not specify what were the parameters that made you choose MSS over gradient (DFI >30% for instance? and what do you mean by the term medical assesment, what does it include?)
It is true, that your data suggest that results are better when performing ICSI using your MSS in cases of paternal age>45. but you rely them upon a poor choice, since you do not provide solid proof upon inclusion and exclusion criteria. You do not have a control group. You should perform MSS and gradient in both cases of high DFI and not, paternal age <45 and >45 and compare the rsults.
I hope that you will find my remarks helpful for your future research.
Reviewer 4 Report
Comments and Suggestions for Authors
Thank you so much for inviting me to review the manuscript “Advanced paternal age: a new indicator for the use of sperm DNA fragmentation selection techniques”.
Please find my review report below.
This is an interesting study that investigated novel indicator for the use of sperm DNA fragmentation selection techniques. Current techniques to study perm DNA fragmentation include sperm chromatin dispersion test, sperm chromatin structure assay, TUNEL and COMET assay. Microfluidic sperm selection devices enable selection of spermatozoa of higher quality and with a lower rate of DNA fragmentation The current study analyzed microfluidic selection devices on outcomes of ART and ICSI. The study investigates fertilization, blastulation, implantation, live-birth and miscarriage rates. IR and LBR are higher in the MSS group while there was no significant difference in MR. The manuscript is well-written, methods are written in detail, and results are clearly described. The results are supported by scientific evidence. I have few comments on the manuscript
1- Ovarian estimulation or Ovarian stimulation?
2- Line 132-136, should be in one paragraph.
3- Figure 1 a, why is r2 but not r on the figure? Also resolution of the figure is a bit low, could be replace with higher resolution image.
4- Please add r, p value on figure 1 b.
5- Please provide details on MSS using the ZyMōtTMICSI device, geometry, design.
6- Does the MSS methods used here inflict any damage on the selected sperm?
Round 2
Reviewer 3 Report
Comments and Suggestions for Authors
Dear authors thank you for your response. I really hope that you found my remarks helpful. best of luck upon your upcoming projects.
Author Response
Thank you very much.